# Antiviral Activity of Catechin against Dengue Virus Infection

**DOI:** 10.3390/v15061377

**Published:** 2023-06-15

**Authors:** Bowen Yi, Benjamin Xuan Zheng Chew, Huixin Chen, Regina Ching Hua Lee, Yuhui Deborah Fong, Wei Xin Chin, Chee Keng Mok, Justin Jang Hann Chu

**Affiliations:** 1Laboratory of Molecular RNA Virology and Antiviral Strategies, Department of Microbiology and Immunology, Yong Loo Lin School of Medicine, National University of Singapore, Singapore 117597, Singapore; bowenyi@nus.edu.sg (B.Y.); benjamin.eric.bc@gmail.com (B.X.Z.C.); micch@nus.edu.sg (H.C.); miclch@nus.edu.sg (R.C.H.L.); micchwx@nus.edu.sg (W.X.C.); 2Infectious Disease Translational Research Programme and Department of Microbiology and Immunology, Yong Loo Lin School of Medicine, National University of Singapore, Singapore 117597, Singapore; 3Biosafety Level 3 Core Facility, Yong Loo Lin School of Medicine, National University of Singapore, Singapore 117599, Singapore; 4A*STAR Infectious Diseases Labs (A*STAR ID Labs), Agency for Science, Technology and Research (A*STAR), Singapore 138648, Singapore; mok_chee_keng@idlabs.a-star.edu.sg; 5Collaborative and Translation Unit for HFMD, Institute of Molecular and Cell Biology, Agency for Science, Technology and Research (A*STAR), Singapore 138673, Singapore

**Keywords:** dengue virus, antiviral, catechin, natural compound

## Abstract

Dengue virus (DENV) is the cause of dengue fever, infecting 390 million people worldwide per year. It is transmitted to humans through the bites of mosquitoes and could potentially develop severe symptoms. In spite of the rising social and economic impact inflicted by the disease on the global population, a conspicuous lack of efficacious therapeutics against DENV still persists. In this study, catechin, a natural polyphenol compound, was evaluated as a DENV infection inhibitor in vitro. Through time-course studies, catechin was shown to inhibit a post-entry stage of the DENV replication cycle. Further investigation revealed its role in affecting viral protein translation. Catechin inhibited the replication of all four DENV serotypes and chikungunya virus (CHIKV). Together, these results demonstrate the ability of catechin to inhibit DENV replication, hinting at its potential to be used as a starting scaffold for further development of antivirals against DENV infection.

## 1. Introduction

Dengue virus (DENV) is an enveloped single-stranded RNA virus belonging to the *Flaviviridae* family of viruses. DENV consists of four serotypes, namely DENV-1, DENV-2, DENV-3, and DENV-4, distinguished by antigenic differences in the structural envelope protein [1]. DENV transmits primarily through the bites of infected female *Aedes aegypti* and *Aedes albopictus* mosquitoes [2]. DENV infection results in dengue fever, which is characterized by an incubation period of 3–14 days and an illness period of 2–7 days. Symptoms of the disease include headache, muscle and joint pain, as well as macular or maculopapular rash. Although most patients recover without medical intervention, severe forms of the disease can occur, resulting in shock or haemorrhage [3].

The incidence of DENV has increased dramatically in over 100 countries worldwide. According to a recent study, the global annual incidence of DENV infections is estimated to be 390 million cases. Among the 3.9 billion individuals at risk for DENV infection, the burden of disease is concentrated mostly in Asia, accounting for approximately 70% of the disease burden [4]. Despite efforts to develop effective drugs to combat DENV, such as balapiravir [5], chloroquine [6], celgosivir [7], lovastatin [8], and prednisolone [9], which have all entered clinical trials for the treatment of DENV infection, none of them have showed sufficient efficacy to be authorized as an antiviral for DENV. Therefore, the development of a novel antiviral agent to effectively combat DENV infection is a pressing and indispensable need, especially in tropical and subtropical countries.

Flavonoids are natural phytochemicals with reported antiviral activities targeting different stages of viral infection [10]. Some flavonoids have been found to be effective against DENV infection, such as betulinic acid [11], ST081006 [12], quercetin [13], and baicalin [14]. We investigated the antiviral activity of catechin, a natural polyphenolic compound belonging to the flavonoid family. Catechin is commonly found in food [15] and plant leaves [16], and has garnered interest as an antiviral agent [17] due to its inhibitory effect against influenza virus [18], CHIKV [19], hepatitis C virus [20], and enterovirus A71 [21]. Hence, we explored the antiviral effect of catechin against DENV replication in vitro. Our findings indicate that catechin treatment resulted in a reduction of viral titres as measured via viral plaque assays. Time-course experiments demonstrated that catechin did not interfere with viral entry, and that its continued presence was required to prevent viral titre recovery from successive replication cycles. Western blot analysis confirmed that catechin inhibited viral protein synthesis. Additionally, the use of a translation reporter assay revealed that catechin inhibited DENV replication through hindering viral protein translation. Furthermore, catechin was able to inhibit all four DENV serotypes and CHIKV, strongly suggesting its potential to be an antiviral agent for DENV infection.

## 2. Materials and Methods

### 2.1. Cell Lines and Viruses

Human hepatoma cells (HUH 7; Dr Priscilla Yang, Harvard Medical School, Boston, MA, USA), human lymphoblast cells (K562; ATCC CCL-243TM), baby hamster kidney (BHK-21; ATCC), and *Aedes albopictus* larvae cells (C6/36; ATCC CRL-1660) were utilised in this study. HUH 7 and K562 cells were cultured in Dulbecco’s Modified Eagle media (DMEM; Gibco, New York, USA) supplemented with 10% heat-inactivated fetal calf serum (FCS; Capricorn Scientific, Ebsdorfergrund, Germany). Baby hamster kidney cells were grown in RPMI-1640 media (Sigma-Aldrich Corp., St Louis, MO, USA) supplemented with 10% FCS. C6/36 cells were cultured in L-15 media (Sigma-Aldrich) containing 10% FCS. C6/36 cells were cultured at 28 °C without CO_2_ while BHK-21, HUH 7 and K562 cells were cultured at 37 °C with 5% CO_2_. For viruses, DENV-1 (EDEN 2928), DENV-2 (16681), DENV-3 (EDEN 2930), DENV-4 (TCR310A129) and chikungunya virus (CHIKV-6708, LK(EH)CH6708, Accession No.: EU441882.1) were used in this study. Propagation of all viruses was accomplished through inoculating confluent C6/36 cells followed by incubation in L-15 media supplemented with 2% FCS. The supernatant was harvested and preserved at a temperature of −80 °C before quantification through viral plaque assay.

### 2.2. Compounds

Catechin (ChemFaces, Wuhan, China) was resuspended in dimethyl sulfoxide (DMSO) to a stock concentration of 100 mM and stored at −80 °C.

### 2.3. Cell Viability and Dose-Dependent Inhibition Studies

HUH 7 cells were seeded on a 96-well plate and incubated at 37 °C with 5% CO_2_ overnight. The cells were then treated with varying concentrations of catechin and incubated for 48 h. After incubation, media in the wells were replaced with alamarBlue™ (Thermo Fisher Scientific, Waltham, MA, USA) diluted in DMEM, supplemented with 2% FCS at a 1:10 dilution, and the 96-well plate was incubated for another 2.5 h at 37 °C, 5% CO_2_. Infinite 200 Pro multiplate reader (Tecan, Männedorf, Zürich, Switzerland) was used to obtain fluorescence readings at an excitation wavelength of 570 nm and an emission wavelength of 585 nm. Relative cell viability was evaluated through normalizing the fluorescence readings of catechin-treated cells to 0.1% DMSO-treated cells.

Assays of dose-dependent inhibition were carried out to determine the inhibitory effects of catechin on virus replication. HUH 7 cells were seeded, left overnight, and subsequently infected with DENV for 1 h at 37 °C, 5% CO_2_. For CHIKV, the infection duration was 1.5 h. Following infection, cells were washed one time with 1× PBS and exposed to 25 µM, 50 µM, or 100 µM catechin suitably diluted in DMEM with 2% FCS, or RPMI with 2% FCS, and incubated at 37 °C, 5% CO_2_ for 48 h. The supernatants were harvested following incubation and viral plaque assays were performed to quantify viral titres.

### 2.4. Viral Plaque Assay

BHK-21 cells were seeded on 24-well plates in RPMI-1640 with 10% FCS and incubated at 37 °C with 5% CO_2_ overnight. BHK-21 cells were infected with tenfold serially diluted samples of virus, which was followed by 1 h incubation at 37 °C with 5% CO_2_. After 1 h, all wells were washed with 1 × PBS two times and 1% carboxymethyl cellulose (CMC) in RPMI-1640 was added to each well. DENV samples were incubated at 37 °C with 5% CO_2_ for six days while CHIKV samples were incubated for two days. Next, the media in each well was decanted and 1 mL of crystal violet with 4% paraformaldehyde staining solution was added. The plates were incubated at room temperature overnight. The next day, the staining solution was aspirated and plaque numbers were counted.

### 2.5. Time-of-Addition and Time-of-Removal Studies

HUH 7 cells were seeded and infected with DENV-2 at a multiplicity of infection (MOI) of 1 for 1 h, after which 1 × PBS was added to the cells for washing. At 0 h post infection (hpi), only DMEM with 2% FCS was added for the time-of-addition (TOA) experiments. At the time points of 0, 2, 4, 6, 12, 18, 24, and 36 hpi, media was aspirated and a concentration of 100 μM catechin was added. Infected HUH 7 cells were incubated with 100 μM catechin at 0 hpi for time-of-removal (TOR) studies. Following the same timepoints as the TOA studies, media containing catechin was decanted and DMEM containing 2% FCS was added to the wells. 0.1% DMSO was used as the vehicle control for both TOA and TOR assays. At 36 hpi, the supernatants were harvested for viral titre quantification.

### 2.6. Pre-Treatment, Co-Treatment and Entry Bypass Studies

To conduct these three investigations, 24-well plates were utilized to culture HUH 7 cells. To conduct the pre-treatment assay, HUH 7 cells were initially incubated with 25 µM, 50 µM, or 100 µM catechin for a duration of 2 h at 37 °C with 5% CO_2_, after which the treated cells were thoroughly rinsed twice with 1 × PBS. HUH 7 cells were subsequently subjected to infection with DENV-2 at MOI 1 and were incubated for a period of 1 h at 37 °C with 5% CO_2_. Post-incubation, 1 × PBS was added to the wells, decanted, and the wells filled with DMEM containing 2% FCS.

The co-treatment study was initiated through incubating DENV-2 with 100 μM catechin at 37 °C for 30 min. Following incubation, the suspension of virus and drug was subjected to centrifugal filtration in a centrifugal filter unit (Sartorius, Göttingen, Germany) at 1500× *g* for 5 min at 4 °C. This step removes unattached drugs. For resuspension, 1 mL of 1 × PBS was added to the suspension and passed through the centrifugal filter unit at 1500× *g* for 5 min at 4 °C. Subsequently, HUH 7 cells were infected with the filtered virus in DMEM with 2% FCS for 1 h at 37 °C with 5% CO_2_. Post-incubation, cells were rinsed with 1 × PBS and replenished with DMEM containing 2% FCS.

The entry bypass study was initiated through extracting viral RNA from supernatant of infected C6/36 cells via the RNeasy Mini Kit (QIAGEN, Hilden, Germany) following the protocol of the manufacturers. Next, HUH 7 cells were transfected with viral RNA and exposed to 25 µM, 50 µM, or 100 µM catechin. The transfection process involved preparing 500 ng of viral RNA in 50 μL of DMEM containing 2% FCS and 1 μL of DharmaFECT 1 (Dharmacon, Lafayette, LA, USA) per well. These solutions were mixed and incubated for 30 min at ambient temperature. After incubation, 100 μL of the complexes were added to HUH 7 cells and the plate was incubated for 1 h at 37 °C with 5% CO_2_. Following transfection, wells were topped up with various concentrations of catechin diluted in DMEM containing 2% FCS.

For all of the above studies, supernatants were collected 48 hpi and catechin treatment. Additionally, 0.1% DMSO served as the vehicle control. Viral titres were then evaluated via viral plaque assays.

### 2.7. SDS-PAGE and Western Blot Analysis

HUH 7 cells seeded on 24-well plates were left overnight. The next day, cells were infected with DENV-2 at an MOI of 1. After 1 h incubation at 37 °C with 5% CO_2_, 1 × PBS was added to the cells for washing, and the cells were treated with varying concentrations of catechin. Cells were lysed with 1 × Laemmli SDS-PAGE buffer at 48 hpi. SDS-PAGE was performed at 100 V for 3 h with a 10% acrylamide gel to separate proteins present in the cell lysate samples. Following gel electrophoresis, proteins were transferred to a polyvinylidene difluoride (PVDF) membrane through a semidry transfer system (Bio-Rad, Hercules, CA, USA) at 1.3 A for 10 min.

2% bovine serum albumin (BSA) with Tris-buffered saline-Tween 20 (TBST) was used to block the PVDF membrane for 1 h before detecting proteins. The primary antibody used was the anti-DENV capsid protein antibody (GTX124247, Genetex, Irvine, CA, USA, 1:1000 dilution), and the membrane was incubated at 4 °C overnight. The next day, TBST was added to the membrane for washing, and each wash lasted 5 min. The secondary antibody used was the polyclonal goat anti-rabbit IgG (H + L) horseradish peroxidase (Thermo Fisher Scientific, 1:5000 dilution), and the membrane was incubated at ambient temperature for 1 h. To detect protein bands, the membrane was rinsed thrice with TBST and developed using the SuperSignal West Pico chemiluminescent substrate (Thermo Fisher Scientific) via the enhanced chemiluminescence (ECL) method. β-actin was used as the loading control through incubating the membrane with anti-β-actin mouse monoclonal 1° antibody (Merck-Millipore, Burlington, NC, USA, 1:10,000 dilution) and AP-conjugated goat anti-mouse IgG 2° antibody (Thermo Fisher Scientific, 1:10,000 dilution) for 30 min.

### 2.8. Translation Reporter Assay

The translation reporter assay was performed using a previously reported replication-defective DENV-2 clone referred to as the NLuc translation reporter construct [11]. The translation reporter genome encodes a nanoluciferase (NLuc) gene, whereby NLuc activity is used as a reporter signal for viral protein translation efficiency [11]. The genome has a deletion in the GDD catalytic triad of the NS5 RNA-dependent RNA-polymerase (RDRP) that renders the RDRP non-functional.

The assay was performed as previously described [11]. HUH 7 cells were first seeded on 24-well plates and incubated overnight at 37 °C with 5% CO_2_. One day post-incubation, the cells were transfected with the NLuc translation reporter plasmid using jetPRIME reagent (Polyplus Transfection, Illkirch-Graffenstaden, France), following the manufacturer’s protocol. Each well was transfected with a transfection mixture containing 50 μL of jetPRIME buffer, 100 ng of accessory plasmid (pTet-Off-Advanced, Clontech, Mountain View, CA, USA), 400 ng of NLuc translation reporter plasmid, and 1 μL of jetPRIME transfection reagent. The cell culture medium and transfection mixture were removed 6 h after transfection and replaced with DMEM containing 2% FCS. The replacement DMEM was also supplemented with 100 μM catechin, 2.5 μM emetine, or 0.1% DMSO control to perform drug treatment. At 72 h post-transfection, NanoLuc^®^ Luciferase assay was carried out as previously described [11] using the Nano-Glo^®^ Luciferase Assay System (Promega, Madison, WI, USA), following the manufacturer’s protocol.

### 2.9. Statistical Analysis

To assess statistically significance in the data obtained, GraphPad Prism (GraphPad Software version 8) was used. One-way analysis of variance (ANOVA) was conducted to determine presence of significant (*p* < 0.05) differences seen in the obtained data. Dunnett’s post-test was conducted on drug-treated samples to compare with control samples. A four-parameter logistic model was used to calculate IC50 and CC50 vales. In addition, paired Student’s *t*-test was carried out to elucidate statistically significant differences between two samples.

## 3. Results

### 3.1. Antiviral Effects of Catechin against DENV-2 Infection In Vitro

HUH 7 cells were treated with catechin at concentrations up to 100 μM for 48 h. Relative cell viability levels remained over 80%, indicating that the tested catechin concentrations are minimally cytotoxic. Following catechin incubation with DENV-2-infected cells, a reduction in viral titres in a dose-dependent manner was observed (Figure 1a). The IC50 value was determined to be 6.422 μM. Further validation procedures were performed on another human cell line, K562, to ensure that the inhibitory effects of catechin were not specific to a particular cell line (Figure 1b). All tested concentrations of catechin showed minimal cytotoxicity in K562 cells and also showed catechin inhibition of DENV-2 in a dose-dependent manner. The values of CC50, IC50, and selectivity index (SI) for both cell lines are calculated in Table 1.

### 3.2. Inhibitory Effect of Catechin on DENV-2 Replication at a Post-Entry Stage

Various time course assays were employed to study the mechanism of catechin inhibition against DENV replication. The time-of-removal (TOR) and time-of-addition (TOA) experiments were performed to investigate whether catechin targets a pre- or post-entry stage in the DENV-2 replication cycle. The TOA assay was performed via introducing 100 µM catechin to infected HUH 7 cells at particular timepoints (0, 2, 4, 6, 12, 18, 24, and 36 hpi), whereas in the TOR study, 100 µM catechin was removed from treated cells at the same timepoints (Figure 2a). For both studies, 0.1% DMSO was used as the vehicle control. The TOA study showed that adding catechin prior to 6 hpi was essential in order to reduce viral titres at 36 hpi. This finding is further evidenced in the TOR study, where removal of catechin only at 6 hpi or later could maintain viral titres at low levels. These results reveal that the inhibitory effects of catechin were present from 6 hpi onwards, leading to viral titre reduction at 36 hpi. This suggests that catechin acts on the DENV-2 replication cycle at the post-entry stage.

Pre-treatment studies were conducted to assess if catechin can affect the entry processes of DENV-2 into HUH 7 cells. However, no significant antiviral effects were observed in comparison to cells pre-treated with 0.1% DMSO, indicating that catechin does not impede virus entry into cells (Figure 2b).

The potential of catechin to prevent virus particle binding to HUH 7 cells was also investigated through examining its effects on viral entry through co-treatment studies. The virus was treated with 100 μM catechin and filtered prior to infection of host cells. Results obtained from the co-treatment studies were similar to those of the pre-treatment studies (Figure 2c), indicating that catechin does not exhibit its antiviral effects via preventing DENV-2 virus particle binding to host cells.

Entry bypass assays were employed to confirm the former findings that catechin affects a post-entry stage of virus replication. Specifically, HUH 7 cells were transfected with viral RNA, bypassing the early replication cycle stages of viral entry and uncoating. Transfected cells were then treated with 25 µM, 50 µM, or 100 µM of catechin. The results show that viral titres decrease dose-dependently (Figure 2d), suggesting that catechin does target a post-entry stage of the DENV-2 replication cycle. Moreover, these findings are consistent with previous TOA and TOR assays.

### 3.3. Inhibitory Effect of Catechin on DENV-2 Viral Protein Translation

Given that catechin inhibits DENV-2 at a post-entry stage of the DENV-2 replication cycle, we investigated whether it has any effect on the subsequent steps of the DENV-2 replication cycle. Specifically, we investigated if catechin had any effect at the pre-packaging stage. This includes the first round of viral protein translation, followed by exponential rounds of viral RNA replication and protein translation. To do this, HUH 7 cells were infected with DENV-2 and then treated with catechin at various concentrations or with 0.1% DMSO control. At 48 hpi, cell lysates were harvested and Western blot was performed to analyse DENV-2 capsid protein levels. It was found that catechin treatment could reduce DENV-2 capsid protein levels (Figure 3a,b). However, the Western blot analysis cannot determine which specific stage of the virus replication cycle is being inhibited and resulting in this decrease in viral protein levels.

Therefore, it was investigated if catechin can specifically inhibit the first round of viral protein translation. A viral translation reporter assay was performed using our previously reported replication-defective DENV clone called the translation reporter construct (Figure 3c) [11]. This translation reporter construct is designed to determine if catechin can specifically inhibit the first round of viral protein translation. First, the translation reporter genome is replication defective because of a deletion in the GDD catalytic triad of the NS5 RDRP (Figure 3c). This GDD deletion does not affect the initial viral protein translation step. Instead, the GDD deletion renders the RDRP non-functional. Second, the genome carries a NLuc reporter gene. Therefore, the viral translation reporter genome retains the ability to undergo the first round of viral protein translation, with the NLuc activity acting as a reporter for protein translation activity. However, the GDD deletion prevents the RDRP from initiating the subsequent RNA replication stage of the replication cycle. Third, the viral translation reporter assay is performed through transfecting the translation reporter plasmid clone into the host cell. After transfection, the host nuclear machinery transcribes the viral RNA genome from the plasmid. The translation reporter genome is then exported directly into the host cytoplasm, where it can directly initiate the first round of viral protein translation. This bypasses the viral particle binding, viral particle entry, and viral particle uncoating steps that take place earlier in the virus replication cycle.

The NLuc translation reporter assay was performed as previously described [11]. HUH 7 cells were transfected with the translation reporter plasmid and treated with 100 μM of catechin. In addition, 0.1% DMSO treatment was a negative control and 2.5 μM emetine treatment was a positive control. Compared to 0.1% DMSO treatment, catechin treatment resulted in an 86% reduction in the NLuc signal (Figure 3d). This demonstrates that catechin might inhibit the first round of viral protein translation. However, further studies are required to investigate and confirm this conclusion.

### 3.4. Antiviral Effects of Catechin against Other Dengue Serotypes and Mosquito-Borne RNA Virus

The antiviral effects of catechin against DENV-1, DENV-3, and DENV-4 replication were tested as well. Catechin demonstrated a dose-dependent reduction in viral titres against these three dengue serotypes at concentrations above 25 μM (Figure 4a–c). Further testing against another mosquito-borne RNA virus, CHIKV, showed that at 100 μM, catechin was effective against CHIKV (Figure 4d). This suggests that catechin is also a potential antiviral for other RNA viruses. The CC50, IC50, and SI values for each virus are calculated in Table 2.

## 4. Discussion

Over the last few years, the incidence of DENV infection has been on the rise in many areas, posing a serious risk to public health worldwide and causing significant social and economic consequences globally. The absence of an FDA-approved therapeutic further emphasizes the urgent need to develop an effective antiviral for DENV [22].

For centuries, medicinal plants have been utilised for the prevention and treatment of various diseases. Flavonoids, a significant subclass of polyphenols, are present in various food and plant species. Due to their diverse pharmacological activities, including antiviral, anti-bacterial, antioxidant, and anti-inflammatory properties, a growing number of studies have been carried out on this class of compounds [12,23,24,25,26]. Catechins are flavonoid derivatives present in food and medicinal plants. Research has shown that consumption of catechin-rich foods may have preventive and therapeutic effects on chronic diseases [27,28] and influenza virus infection in humans [29,30]. Previous studies have evaluated the antiviral activity of catechins against DENV-2 through molecular docking [31,32]. Another major catechin compound found in green tea, epigallocatechin-3-gallate (EGCG), has showed inhibitory effects against DENV [33,34] and hepatitis C virus infection [35]. This study explored the potential therapeutic application of catechin against DENV infection. We evaluated the dose-dependent inhibitory effects of catechin on DENV-2 in vitro. Infected HUH 7 cells treated with catechin at non-cytotoxic concentrations showed significant decrease in viral titres with an IC50 value of 6.422 μM. This effect of catechin was also observed in another DENV-permissible cell line, K562.

To elucidate the replication cycle stages targeted by catechin, several time-course studies were conducted. The TOA study showed that addition of catechin after 6 hpi resulted in higher viral titres as compared to the earlier timepoints. This suggests that at 6 hpi, catechin failed to fully inhibit DENV infection as some virus replication has already occurred. Whereas the TOR study revealed that catechin removal before 6 hpi resulted in higher viral titres as compared to timepoints after 6 hpi. This indicates that the continued presence of catechin is required for the continued inhibition of DENV-2 replication, while removing catechin early may lead to viral titre recovery from successive replication cycles. Pre- and co-treatment assays also point to the activity of catechin to be post-entry as catechin failed to induce any inhibitory effects on viral entry. Additionally, the entry bypass assay demonstrated catechin still exhibited its inhibitory effects on DENV-2 in the absence of the viral entry stages. Altogether, these findings suggest that catechin is unlikely to affect viral entry but instead inhibits DENV-2 replication post-entry.

Results indicate that catechin might inhibit viral protein translation. Western blot analysis shows that there is inhibition of DENV-2 capsid protein expression (Figure 3a,b). It was then investigated if catechin could specifically inhibit the first round of viral protein translation. To do this, a viral translation assay was performed with a replication-defective viral translation reporter construct. This viral translation reporter assay is designed to determine if catechin can specifically inhibit the first round of viral protein translation. The translation reporter assay does so through bypassing all earlier stages of the virus infection cycle and through blocking progress to all subsequent steps of the virus infection cycle. This does not exclude the possibility that catechin can target other stages of the virus replication cycle. For example, the subsequent exponential stages of viral RNA replication and viral protein translation are tightly coupled together. Since viral protein expression is already inhibited, less DENV-2 NS5 RDRP will be produced, which will result in lower levels of RNA replication. Therefore, it will be difficult to determine if catechin has the additional ability to inhibit DENV-2 NS5 RDRP activity. Nonetheless, our translation assay demonstrates that catechin might have the ability to inhibit early viral protein translation (Figure 3c,d). However, further research is required to provide more evidence to support this finding.

Catechin inhibited the replication of not only DENV-2, but also the other three DENV serotypes, demonstrating its potential as an antiviral for DENV. Through dose-dependent inhibition studies, catechin-treatment resulted in viral titre reduction in the other three DENV serotypes and also in another mosquito-borne virus, CHIKV. CHIKV belongs to the genus *Alphavirus*, which commonly exists in similar regions as multiple serotypes of DENV [36]. As catechin was also able to inhibit CHIKV infection, it has the potential to function as a broad-spectrum antiviral for mosquito-borne RNA viruses.

Overall, our findings suggest that catechin is a potent inhibitor of DENV, acting specifically at a post-entry stage of the viral replication cycle. Our studies suggest that catechin might affect DENV-2 replication through affecting viral protein synthesis. As demonstrated by its inhibitory effect against all DENV serotypes and CHIKV, catechin is a promising broad-spectrum antiviral candidate. As catechin is derived from food, it is generally safe for human consumption, giving confidence to its safety in vivo. Catechin production also has lesser impact on the environment as compared to other drugs, adding to its advantage to be developed as a clinical drug candidate. However, further investigations are necessary to gain a better comprehension of its molecular mechanisms, accompanied with in vivo safety and efficacy studies. Nevertheless, considering the risks to global public health posed by DENV, catechin represents an indispensable cornerstone for future therapeutic development.

## Figures and Tables

**Figure 1 viruses-15-01377-f001:**
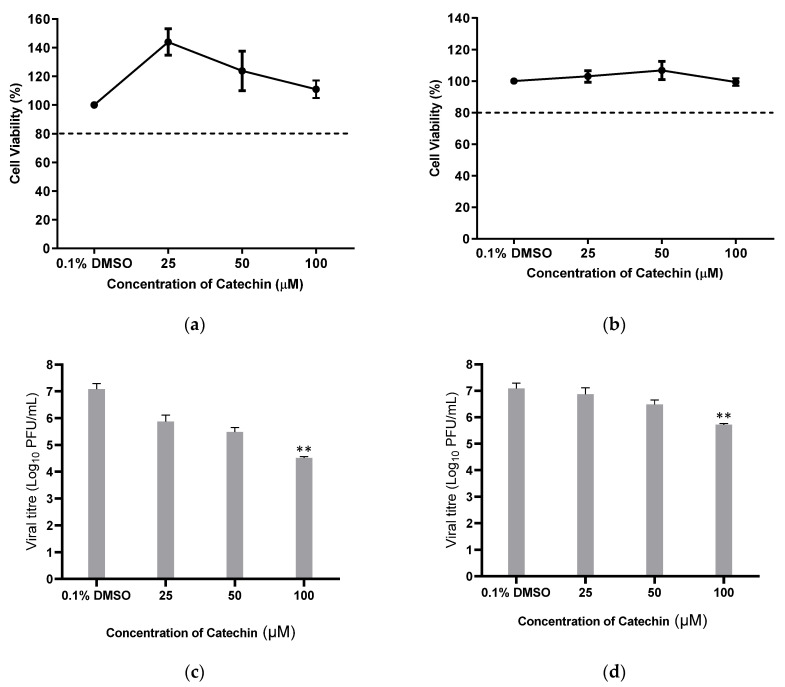
Cell viability of catechin in (**a**) HUH 7 and (**b**) K562 cell lines and antiviral effects of catechin on DENV-2 in (**c**) HUH 7 and (**d**) K562 cell lines. Cells were infected with DENV-2 at MOI 1 and treated with catechin at various concentrations. The dashed line in (**a**,**b**) represents the CC20 cut-off for cell viability. One-way ANOVA and Dunnett’s post-test were used to determine the presence of any statistical differences, with ** denoting that *p* < 0.01. Error bars represent the standard deviation observed from the mean of triplicates performed for both cell viability and dose-dependent inhibition studies.

**Figure 2 viruses-15-01377-f002:**
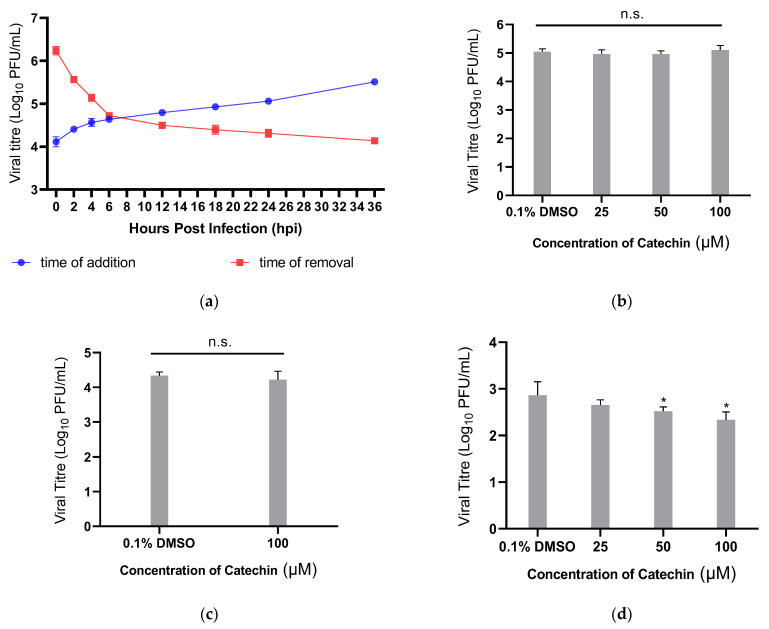
Time-course studies of catechin on DENV-2 infection in HUH 7 cells. (**a**) TOA and TOR studies of catechin. 100 μM catechin was added to or removed from DENV-2-infected HUH 7 cells at specific intervals post-infection. The supernatants were harvested at 36 hpi and viral titres were quantified. (**b**) Pre-treatment studies of catechin. HUH 7 cells were pre-treated with various concentrations of catechin 2 h prior to infection with DENV-2. (**c**) Co-treatment studies of catechin. DENV-2 was incubated with 100 μM catechin for 30 min, after which the suspension was filtered and used to infect HUH 7 cells. (**d**) Entry bypass studies of catechin. HUH 7 cells were transfected with DENV-2 viral RNA, followed by treatment with various concentrations of catechin. For the studies in (**b**–**d**), supernatants were harvested at 48 hpi and viral titres were quantified via plaque assays. One-way ANOVA and Dunnett’s post-test were used to determine the presence of statistically significant difference, with * denoting that *p* < 0.05 and n.s. denoting that *p* > 0.05. Error bars indicate the standard deviation observed from the mean of triplicates that were performed for each study.

**Figure 3 viruses-15-01377-f003:**
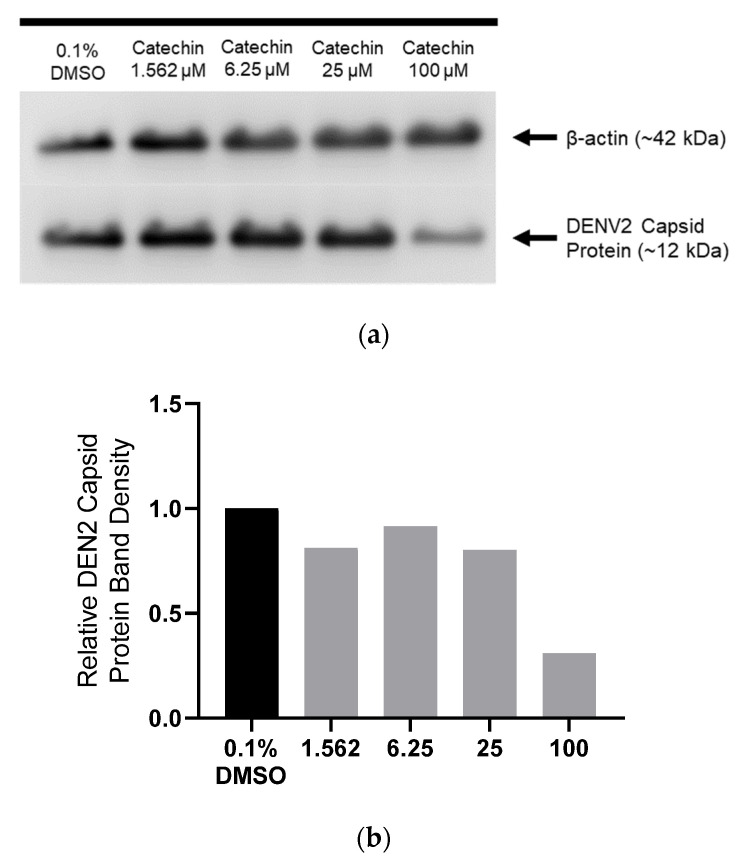
Effects of catechin on DENV-2 viral capsid protein translation. (**a**) Western blot analysis revealed decreasing amount of DENV-2 capsid protein expression after treatment with 1.562, 6.25, 25, and 100 μM catechin, as compared to 0.1% DMSO control. (**b**) Relative band intensities for DENV-2 capsid protein expression normalized to the 0.1% DMSO-treated condition. (**c**) Genomic map of the DENV-2 translation reporter clone used for the translation reporter assay. (**d**) Translation reporter assay showed a decrease in luciferase signals from cells treated with catechin relative to 0.1% DMSO-treated cells, further demonstrating that catechin affects viral protein translation. A concentration of 2.5 μM emetine was used as a positive control. One-way ANOVA and Dunnett’s post-test were used to determine the presence of any statistical differences, with ** denoting that *p* < 0.01. Error bars represent the standard deviation observed from the mean of triplicates performed for the translation reporter assay.

**Figure 4 viruses-15-01377-f004:**
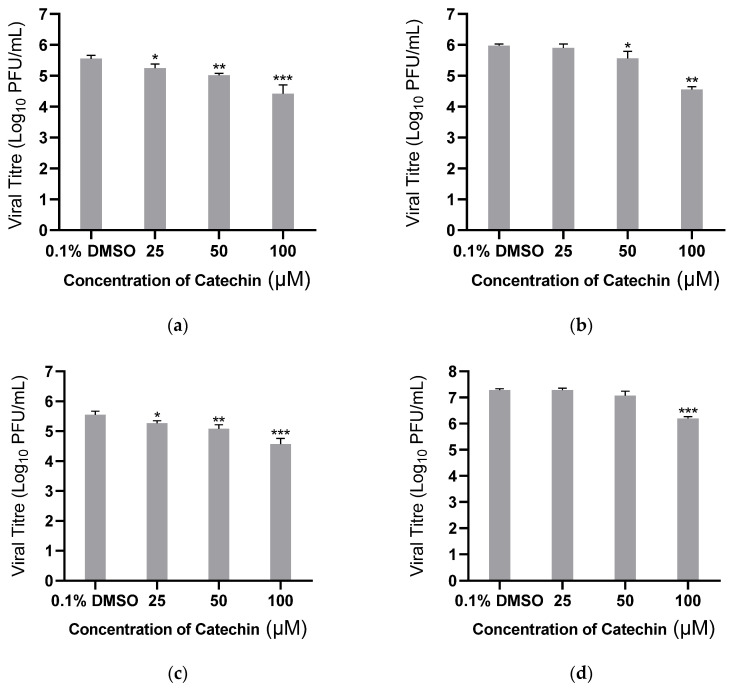
Antiviral effects of catechin on other DENV serotypes and CHIKV. HUH 7 cells were infected with (**a**) DENV-1, (**b**) DENV-3, (**c**) DENV-4, and (**d**) CHIKV at MOI 1 and treated with various concentrations of catechin. Supernatants were harvested for all studies and viral titres were quantified. The primary axis corresponds to viral titre. One-way ANOVA and Dunnett’s post-test were used to determine the presence of any statistical differences, with * denoting that *p* < 0.05, ** denoting that *p* < 0.01, and *** denoting that *p* < 0.001. Error bars represent the standard deviation observed from the mean of triplicates performed the dose-dependent inhibition studies.

**Table 1 viruses-15-01377-t001:** CC50, IC50, and SI values for catechin-treated cells against DENV-2 in HUH 7 and K562 cell lines.

Cell Line	CC50 (μM)	IC50 (μM)	Selectivity Index
HUH 7	>100	6.422	>15.57
K562	>100	50.28	>1.99

Note: CC50 (concentration estimated to reduce the number of viable cells by 50%) and IC50 (concentration estimated to inhibit 50% of the virus) values were derived from data obtained from cell viability and dose-dependent inhibition studies, respectively.

**Table 2 viruses-15-01377-t002:** CC50, IC50, and SI values for catechin-treated cells against the other three DENV serotypes and CHIKV.

Virus	CC50 (μM)	IC50 (μM)	Selectivity Index
DENV-1	>100	37.94	>2.64
DENV-3	>100	38.95	>2.57
DENV-4	>100	16.83	>5.94
CHIKV	>100	47.14	>2.12

Note: CC50 (concentration estimated to reduce the number of viable cells by 50%) and IC50 (concentration estimated to inhibit 50% of the virus) values were derived from data obtained from cell viability and dose-dependent inhibition studies, respectively.

## Data Availability

All data in this study is available in the main text.

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
