# Peer review of "Antiviral Activity of Catechin against Dengue Virus Infection"

_viruses, 2023, doi:10.3390/v15061377_

Round 1

Reviewer 1 Report

Comments and Suggestions for Authors

This manuscript reports on investigations into the antiviral activity of cathechin against dengue virus (DENV) using cell culture models and systems. The authors demonstrate that cathechin treatment inhibits infectious virus production by Huh-7 and K562 cells in a dose-dependent manner, although statistically significant antiviral effects are only observed at high concentrations of the compound (100 uM), where moderate decreases in cell viability are observed. Time-of-addition (TOA) and time-of-removal (TOR) assays are well-executed and well-presented and these studies reveal that the antiviral effects of cathechin are strongest when the compound is present during early stages of infection (up to ~6 hours post-infection). Subsequent experiments demonstrate that pre-treatment and co-treatment with the compound has no significant inhibitory effects on infectious DENV production, while ‘entry bypass’ studies involving transfection of full-length DENV RNA and subsequent treatment with cathechin reveal significant antiviral effects at lower concentrations of cathechin. Antiviral effects on DENV protein translation are then reported, although some additional experimentation and explanation may be required to confirm that these effects are reproducible and specific to viral protein translation. Importantly, the authors demonstrate that the antiviral effects of cathechin are also observed for other DENV serotypes and CHIKV. Overall, the antiviral effects of cathechin against DENV are convincingly demonstrated and investigated using appropriate assays however further studies are required to definitively demonstrate that viral protein translation is the major target of cathechin. The manuscript requires careful editing for English language.

Major Points:

1. The high concentrations of cathechin that are required for statistically significant antiviral effects in most experiments (100 uM; Figs. 1A-B, 4B and 4D) raises some questions as to the potency of compound and the extent to which antiviral effects can be distinguished from its cytotoxic/antiproliferative effects at these higher concentrations. In this context, some additional experiments using a more sensitive viability reagent (e.g. alamar Blue HS or CellTiter Glo2.0 instead of alamar Blue) and inclusion of a positive control for the these assays would add confidence to the reported ‘minimal toxicity’ of cathechin. Also, as error bars are not visible/shown for viability assays the reproducibility of these effects are not completely clear. Finally, it is not clear how EC50 and CC50 values have been calculated as this is not detailed in the Materials and Methods (normally calculated using a 4-parameter logistic model) and it is questionable whether 4-point dose-response assays are sufficient for accurate EC50 and CC50 value determinations.

2. Some of the strongest antiviral effects of catechin are seen in entry bypass assays (Fig. 2D), with concentrations of 3 and 4uM causing significant inhibition of infectious virus production as compared to 25uM-100uM concentrations in other assays. However, no details are provided about the infectious clone that has been used in these assays and it is surprising that the low concentrations of cathechin used in these assays (1-4uM) are not discussed in greater detail.

3. A major conclusion of the study is that cathechin inhibits viral protein translation (Fig. 3). However, it is difficult to ascribe the effects on viral capsid protein levels to translation alone, distinct from early replication (Fig. 3A-B). These Western blotting experiments should ideally be refined to specifically investigate viral protein translation (e.g. following transfection with a replication-defective DENV RNA genome) or conclusions should be tempered. Also, this experiment should be repeated to confirm reproducibility of antiviral effects. Finally, the translation reporter construct should be described in more detail (i.e is it a replication-defective DENV-NLuc reporter genome?), possibly including a schematic diagram of this construct. Also, ideally, similar experiments should be performed using a non-viral translation reporter construct to explore/confirm whether cathechin-mediated impairment of translation is unique to cap-dependent DENV RNA translation.

Minor points

1. It would be helpful if key experimental details (e.g. times, MOIs etc.) were included in Figure Legends as well as Materials and Methods.

2. Details about catechin source, stock preparations, storage etc. should be included.

3. The text should be carefully edited for English grammar, readability and meaning (e.g. line 164 ‘to detect loading control’, line 167 ‘membrane was developed’, line 183 ‘employed for result analysis’, line 186 ‘figure out the significance’, line 196 ‘to assure that’, line 198 ‘shown minimal toxicity’, line 222 ‘fromtreated’ line 225 ‘It necessary’, line 262 ‘the key process of the DENV2 replication cycle’(?), line 339 ‘It demonstrated that’, line 353 ‘unable to process’ etc.) .

4. Care should be taken to ensure consistency in reagent names (e.g. Huh 7, HUH 7) and terminology.

5. Some other studies on the antiviral effects of catechin/EGCG towards DENV and related viruses should probably be acknowledged, discussed and cited (e.g. Raekiansyah M et al Arch Virol 2018, Vazquez-Calvo A et al Front Microbiol, Ciesek S et al Hepatology 2011).

Comments on the Quality of English Language

As detailed above, the manuscript should be carefully edited for English grammar, language and meaning.

Reviewer 2 Report

Comments and Suggestions for Authors

The manuscript entitled "Potential Therapeutic Application of Catechin Against DENV Infection” investigates the antiviral potential of catechin against dengue virus (DENV) and Chikungunya virus (CHIKV). The study demonstrates that catechin, a flavonoid derivative found in foods and medicinal plants, exhibits inhibition of DENV and CHIKV, suggesting its potential use as a broad-spectrum antiviral agent. Indeed, the study further elucidates that catechin targets a post-entry stage in the DENV replication cycle, particularly affecting viral protein synthesis. However, there are several areas in the manuscript that could benefit from clarification and additional detail to enhance the overall quality and impact of the work. The following are major and minor comments for the authors to consider.

Consider the following comments and suggestions for each section of the manuscript:

General

The manuscript is generally understandable, but several instances require editing for clarity, consistency, and academic style.
There are numerous instances where the language could be improved for clarity and academic tone. For example, replace informal terms such as "a few" with more precise language like "several" or "numerous."

In the Results section, the sentence "The antiviral activity of catechin against DENV2 was evaluated through dose-dependent inhibition studies in vitro." could be improved for clarity by specifying what was evaluated, e.g., "We evaluated the dose-dependent inhibitory effects of catechin on DENV2 in vitro."

For example, in the Discussion section, the phrase "In the past a few years" should be revised to "In the past few years". Similarly, the sentence "The urgent need to develop an effective antiviral against DENV is further emphasized, given the absence of an authorized therapeutic." would be clearer if rephrased to "The absence of an authorized therapeutic further emphasizes the urgent need to develop an effective antiviral against DENV."

In various instances, the text lacks the formal tone expected in academic writing. For instance, the phrase "In conclusion, our findings indicate catechin as a strong inhibitor of DENV and affects a stage of post-entry." could be rephrased to a more formal tone, such as "In conclusion, our findings suggest that catechin is a potent inhibitor of DENV, acting specifically at a post-entry stage of the viral replication cycle."

These are just a few examples; a thorough editing pass would enhance the manuscript's clarity and readability. Please consider consulting an experienced editor or proofreader.

Introduction
Minor Comments:

Please ensure that the introduction is concise and directly relevant to the research question. Avoid extraneous details and focus on setting up the rationale for your study.

Major Comments:

The introduction could benefit from a more comprehensive overview of the current state of research on antiviral compounds, specifically catechin. This will put your study into a better context for the readers. References to previous studies on catechin and its antiviral properties would be useful.
It would be beneficial to include a more comprehensive literature review at the beginning of your manuscript, specifically regarding the use of flavonoids as antivirals and the mechanisms of DENV and CHIKV replication. This would provide a stronger foundation for your research and its significance.

Materials and Methods

Major Comments:

The methods section requires more detailed descriptions of the experimental procedures. For example, the dose-dependent inhibition studies should specify the range of catechin concentrations used and the criteria for determining cytotoxicity. The methods for the TOA and TOR assays and the construction of the DENV2 translation reporter clone also need to be elucidated further.

The manuscript does not provide complete information about the statistical tests used to analyze the data. Please specify the statistical tests employed and any assumptions that were checked before performing the tests. Also, include the specific p-values for the results rather than broad descriptors such as "significant." This information would strengthen the credibility of your results and conclusions.

Results

Major Comments:

While the results are generally well presented, the statistical analysis could be strengthened. It is recommended to provide the exact p-values for your tests rather than only providing the IC50 values. This will give the readers a better sense of the significance and robustness of your findings.

Western Blot Analysis (Results): The study used Western blot analysis to investigate the effect of catechin on viral protein translation, and it was analyzed as part of a pixel semi-quantitative method, which produced a bar plot with no error bars because only one WB experiment was performed?

Discussion

Minor Comments:

Please ensure that your conclusions are directly supported by your results. Avoid overreaching or speculative claims.

Major Comments:
The manuscript seems to lean heavily on in vitro studies using only two cell lines, HUH 7 and K562. Although these are commonly used models, the addition of more diverse in vitro models or in vivo studies could strengthen the generalizability of your findings and reduce potential methodological biases.

The discussion could be improved by providing a more in-depth interpretation of the results. How do your findings fit into the broader context of the field? What are the implications for future research or therapeutic development?

I hope these comments will guide you in the revision of the manuscript. Please note that each of these suggestions is intended to help enhance your work's scientific soundness and presentation quality. Because I think your study contributes valuable insights into the field, and with these revisions, it will undoubtedly interest the readers.

Comments on the Quality of English Language

The overall quality of English language used in your manuscript requires improvement to ensure the clarity and comprehensibility of your research. Here are my specific comments:

Throughout the manuscript, there are several instances where sentences could be structured more clearly. For example, in the Discussion section, the phrase "In the past a few years" should be revised to "In the past few years". Similarly, phrases like "posing a crucial threat" and "presenting great social and economic impact" could be rephrased for clarity and simplicity.

In certain parts, the manuscript lacks appropriate transitional phrases, making the text appear disjointed. More cohesive language should be used to enhance the flow of your discussion.

Some sentences are excessively long and convoluted, which can make it difficult for readers to follow your argument. Breaking these sentences into smaller, more manageable units of thought can enhance readability.

The manuscript could benefit from careful proofreading to correct minor grammatical errors, punctuation inconsistencies, and typos. For instance, in the Discussion section, "the urgent need to develop an effective antiviral against DENV is further emphasized" might be better phrased as "the urgent need to develop an effective antiviral for DENV is further emphasized."

Lastly, ensure that technical terms and abbreviations are defined clearly upon first use, and consistently applied throughout the text. This will ensure that your readers, even those who are familiar with the subject matter, can fully comprehend your work.

I recommend seeking assistance from a professional scientific editor or a native English speaker to ensure the manuscript meets a high standard of academic English. This will greatly enhance the readability of your work and ensure that your valuable findings are accessible to a broad scientific audience.

Round 2

Reviewer 1 Report

Comments and Suggestions for Authors

Point 1: It is reassuring that the authors have repeated the cell viability assays in triplicate and this strengthens the conclusion that cathechin does not markedly affect cell viability. However, it is not appropriate that the exact same viability assay data is presented again and again on multiple 'viral titre' graphs (Figs. 1A, 4A, 4B, 4C and 4D). I suggest that this data is presented once only, perhaps as a separate Figure exploring the impact of cathechin on cell viability.

Point 2: The source and purity of the 'DENV-2 viral RNA' that is used in the 'entry bypass assay' is unclear. I can only assume that the authors have transfected cells with RNA that has been extracted from virus preparations (?) which will also include other undefined host RNA species that could complicate interpretations. I suggest that this data is removed or at least these experiments (and in particular the source and purity of the 'DENV-2 viral RNA') are more clearly described.

Point 3: I disagree that data from repeated Western blotting experiments (including means and S.D.)  could not be presented in Fig. 3A. As inhibition of viral protein translation is a major conclusion of the study, I would argue that the Western blotting data/figure is not particularly convincing. The inclusion of the schematic diagram and added description for the 'translation reporter construct' is helpful. I am not entirely convinced that  the effects of cathechin on viral protein translation are specific (and not reflective of a more global effect on RNA translation) and I suggest that conclusions about this antiviral mechanism of cathechin are tempered/carefully qualified.      

Comments on the Quality of English Language

The quality of English language is much improved in the revised manuscript. Basic copy-editing by journal staff should be sufficient to identify and correct any remaining issues. 

Reviewer 2 Report

Comments and Suggestions for Authors

Dear Authors,

I have thoroughly reviewed your revised manuscript and the changes you have made in response to the previous reviewer's comments. Here are my comments and suggestions:

1. The language and academic tone of the manuscript have been significantly improved, enhancing the clarity and readability of the text.

2. The introduction is now concise and directly relevant to the research question. The comprehensive overview of the current state of research on antiviral compounds, specifically catechin, provides a solid context for your study.

3. The detailed descriptions of the experimental procedures in the Materials and Methods section have made the methodology of your study clear and reproducible.

4. Providing the exact p-values for your tests has strengthened the statistical analysis of your results. The inclusion of the pixel semi-quantitative method of the bar plot from the WB experiment has added value to your findings.

5. The discussion section now provides a more in-depth interpretation of the results. The acknowledgment of the limitation of using only two cell lines and the mention of future in vivo studies show your commitment to advancing this research.

6. The improvements in the English language used in the manuscript are evident. The sentences are now structured more clearly, the text flows smoothly with appropriate transitional phrases, and the technical terms and abbreviations are defined clearly upon first use.

Suggestions:

- I recommend a final proofreading of the manuscript to ensure there are no overlooked minor errors or inconsistencies.

- Consider expanding the range of cell lines or models used in your future studies to further validate your findings and enhance the generalizability of the results.

- Continue to provide detailed methodologies and statistical analyses in your future work. This transparency not only strengthens your current study but also contributes to the reproducibility of your research in the scientific community.

Overall, your revised manuscript shows a significant improvement and provides valuable insights into the potential therapeutic application of catechin against DENV infection.

Best regards
